# Osteoporosis from an Endocrine Perspective: The Role of Hormonal Changes in the Elderly

**DOI:** 10.3390/jcm8101564

**Published:** 2019-10-01

**Authors:** Rossella Cannarella, Federica Barbagallo, Rosita A. Condorelli, Antonio Aversa, Sandro La Vignera, Aldo E. Calogero

**Affiliations:** 1Department of Clinical and Experimental Medicine, University of Catania, 95123 Catania, Italy; federica.barbagallo11@gmail.com (F.B.); rosita.condorelli@unict.it (R.A.C.); sandrolavignera@unict.it (S.L.V.); 2Department of Experimental and Clinical Medicine, University Magna Graecia of Catanzaro, 88100 Catanzaro, Italy; aversa@unicz.it

**Keywords:** osteoporosis, bone mineral density, FSH, testosterone, cortisol, IGF1, vitamin D

## Abstract

**Introduction:** Osteoporosis is increasingly prevalent in the elderly, with fractures mostly occurring in women and men who are older than 55 and 65 years of age, respectively. The aim of this review was to examine the evidence regarding the influence of hormones on bone metabolism, followed by clinical data of hormonal changes in the elderly, in the attempt to provide possible poorly explored diagnostic and therapeutic candidate targets for the management of primary osteoporosis in the aging population. **Material and methods:** An extensive Medline search using PubMed, Embase, and Cochrane Library was performed. **Results:** While the rise in Thyroid-stimulating hormone (TSH) levels has a protective role on bone mass, the decline of estrogen, testosterone, Insulin-like growth factor 1 (IGF1), and vitamin D and the rise of cortisol, parathyroid hormone, and follicle-stimulating hormone (FSH) favor bone loss in the elderly. Particularly, the AA rs6166 FSH receptor (FSHR) genotype, encoding for a more sensitive FSHR than that encoded by the GG one, is associated with low total body mass density (BMD), independently of circulating estrogen. A polyclonal antibody with a FSHR-binding sequence against the β-subunit of murine FSH seems to be effective in ameliorating bone loss in ovariectomized mice. **Conclusions:** A complete hormonal assessment should be completed for both women and men during bone loss evaluation. Novel possible diagnostic and therapeutic tools might be developed for the management of male and female osteoporosis.

## 1. Introduction

Osteoporosis was firstly defined by the International Consensus in 1993 as a systemic skeletal disease of which its main features are low bone mass and microarchitectural deterioration of the bone tissue, leading to bone frailty and fracture susceptibility [1]. According to the World Health Organization (WHO) criteria, a bone mineral density (BMD) T-score of 2.5 or less indicates osteoporosis, whereas osteopenia is defined for values ranging between −1 and −2.5 [2].

The skeleton renews every 10 years through a process called bone remodeling by which the old bone is replaced by the new one. This happens in the bone remodeling units where, in well-defined temporally and spatially-coupled events, osteoclasts are firstly recruited to reabsorb a quantum of mineralized bone and then undergo apoptosis. Osteoblasts are in turn recruited in the same site to make and mineralize new bone tissue [3,4]. In the young adult human skeleton, there is a quantitative balance between the amount of bone formation and resorption. In contrast to bone remodeling, bone resorption and formation are not temporally or spatially coupled in bone modelling, a process that is important for bone growth and development, occurs after mechanical stress, and is aimed at sculpting bones and optimizing their shape and structure [5,6]. Each event that is able to reduce bone formation or increase bone resorption can lower the BMD.

Osteoporosis-associated fractures are increasingly prevalent in women after 55 years of age, as well as in men after 65 years of age, indicating the negative impact of aging on bone metabolism. Aging affects the remodeling balance in a sex-specific manner. In women, it is associated with increased bone reabsorption and in men, it is associated with decreased bone formation and turnover [7]. The influence of hormones on both osteoblast and osteoclast metabolism is not limited to sex hormones, as pre-clinical evidence shows, however several other hormones, of which their levels vary with age, are able to affect it. The aim of this review is to summarize the basic evidence on the influence of hormones on bone metabolism, followed by clinical data of hormonal changes during aging, in the attempt to provide possible poorly explored diagnostic and therapeutic candidate targets of osteoporosis in the elderly. To accomplish this, an extensive search in PubMed, Embase, and Cochrane Library was performed by two independent authors using the following key-words: “osteoporosis”, “bone mineral density”, “Thyroid-stimulating hormone (TSH)”, “cortisol”, “estradiol”, “testosterone”, “follicle-stimulating hormone (FSH)”, “luteinizing-hormone (LH)”, parathyroid hormone”, “vitamin D”, and “Insulin-like growth factor 1 (IGF1)”. Only English language studies that were published from each database’s inception to 30 July 2019 were included. In addition, the reference lists from the articles were searched. There was no restriction regarding the design that the study’s used. 

## 2. Pre-Clinical Evidence

The equilibrium between bone reabsorption and bone formation during bone remodeling is due to the balanced activity between osteoclasts and osteoblasts. This balance is mostly regulated by the receptor activator of nuclear factor-κB ligand (RANKL)/RANK (its receptor)/osteoprotegerin (OPG) (triggering osteoclastogenesis) and the Wnt/β-catenin (triggering osteoblastogenesis) pathways [8]. Several cytokines have also been recognized as playing a role in such a balance, since pro-inflammatory ones (e.g., IL-1, IL-6, TNFα), released by osteoblasts and T-cells, may accelerate bone reabsorption [9,10]. Several hormones have been found to modulate RANKL/RANK/OPG, Wnt/β-catenin pathways or cytokine release. The role of thyroid hormones, glucocorticoids (GCs), sex hormones, gonadotropins, parathyroid hormone, vitamin D, and insulin-like growth factor 1 (IGF1) on osteoblast and osteoclast metabolism is discussed and summarized in Table 1.

### 2.1. Thyroid Hormones and Thyroid-Stimulating Hormone

Thyrotoxicosis is associated with a temporal uncoupling of bone remodeling, leading to bone loss [11]. This was firstly addressed by early in-vitro studies showing the impact of thyroid hormones (T3 and T4) on bone resorption and formation [12,13,14]. However, bone remodeling is not affected in mice who lack thyroid hormone receptors (TR) α1/β [15], thus indicating that thyroid hormones might not directly influence bone metabolism. Thyroid-stimulating hormone (TSH) receptor (TSHR) is expressed in both osteoclast and osteoblast precursors. A 50% decrease in TSHR expression causes osteoporosis and focal osteosclerosis in euthyroid null mice. In addition, TSH has been found to hinder osteoclast formation and survival acting on the JNC/c-jun and NFkB signaling triggered by RANKL and TNFα, as data from osteoclast or osteoblast cultures incubated with rhTSH indicate. It also inhibits osteoclast differentiation down-regulating Wnt and VEGF signaling [16]. These data support the hypothesis of a direct effect of TSH on bone metabolism. Particularly, TSH directly decreases bone remodeling, acting both on osteoclast formation and survival and on osteoblast differentiation [16].

### 2.2. Glucocorticoids and Adrenocorticotropin Hormone

GCs influence bone metabolism primarily acting on osteoblasts, since osteoclast metabolism does not seem to be affected by GCs [17]. Accordingly, the inhibition of bone formation has been suggested to be the major contributor in GC-induced osteoporosis. An in-vitro study found that dexamethasone, over the dose of 10^–8^ M, inhibited osteogenesis and osteogenic protein expression (e.g., alkaline phosphatase (ALT) and osteocalcin (OCN)) by down-regulating the PI3K/Akt pathway phosphorylation [18]. GCs can reduce osteoblast maturation, lifespan, and function. Furthermore, GCs can cause osteonecrosis or avascular necrosis by inducing osteoblast apoptosis and decreasing bone blood flow and hydration [19,20]. Immuno-histochemical evaluation found that osteocytes, which are terminally differentiated osteoblasts lying in the bone matrix and connecting to other osteoclasts, blood vessels, and bone surface also undergo autophagy or apoptosis following GC exposure. Indeed, after GC incubation, they start to secrete cathepsin K, which removes type I collagen and other proteins from the bone matrix, thus promoting the increase in size of osteocyte lacunae [19].

Conversely, the Adrenocorticotropin hormone (ACTH) has shown to have protective effects against GC-induced osteonecrosis [21]. A study assessed whether incubation with the ACTH may influence human osteoblasts differentiation. At the dose of 10 pM, ACTH exposure resulted in the enhancement of osteogenesis by accelerating the expression of bone-specific genes (e.g., collagen I, biglycan, the vitamin D receptor, and TGF-β) [22]. Such findings have also been confirmed elsewhere [23].

### 2.3. Sex Hormones and Gonadotropins

Estrogens have long been considered as inhibitors of bone resorption, hindering osteoclast differentiation. In-vitro evidence supports a modulating role for estrogens on the release of inflammatory cytokines (IL-1, IL-6, TNFα) from osteoblasts and T-cells [9,10]. They could also hamper osteoclast differentiation by acting on bone-marrow precursors [24,25]. The underlying molecular mechanisms are not entirely clear, however they mainly derive from mouse models with a specific deletion of estrogen receptor (ER) in selected bone cells. Indeed, the selective ERα knockout in osteoclasts from female mice (but not in males) causes post-menopausal-like osteoporosis. Estrogens seem to directly regulate osteoclast’s lifespan by the Fas/FasL system [26]. Similarly, deletion of ERα in osteoblast progenitors lead to a reduction in periostal bone apposition, lowering cortical bone mass. This is thought to have resulted from the estrogen-dependent Wnt/β-catenin pathway enhancement, which in turn leads to the proliferation and differentiation of osteoblast progenitor cells [27]. In addition, the OPG/RANKL system has been suspected as one of the downstream mediators of the ERα-downstream pathway [28].

Androgens can influence bone metabolism either directly, binding their specific receptor (AR), or indirectly, by acting on ER after their aromatization. The AR has been identified in cultured human fetal osteoblasts where it promotes cell proliferation and differentiation by inhibiting their apoptosis via IL-1β and FGF-mediated effects [29]. Androgens also seem to indirectly suppress osteoclast proliferation, since hypogonadism consequent to orchiectomy leads to osteocyte proliferation. Indeed, hypotestosteronemia rises RANKL secretion by osteoblast precursors, thus, in turn, stimulating osteoclast proliferation [29]. 

Some authors have reported the occurrence of only mild osteopenia in ERα or ERβ null mice and normal bone mass in mice who are null for both receptors [30,31,32]. Furthermore, bone loss has been reported in ovariectomized mice, but not in mice undergoing ovariectomy plus hypophysectomy [33,34]. These observations led to speculation of a possible role for follicle-stimulating hormone (FSH) in bone metabolism. FSH receptor (FSHR) and FSHβ knockout ovariectomized mice do not develop bone loss, suggesting that the FSH/FSHR pathway is involved in the pathogenesis of osteoporosis. In-vitro experiments showed that FSH stimulates osteoclasts in a Gi2α-coupled FSHR manner, triggering the MEK/Erk, NFkB, and Akt pathways, enhancing osteoclastogenesis and bone resorption [35]. A 13-aminoacid-long peptide polyclonal antibody that is capable of selectively binding and inhibiting the FSHβ subunit has been used in animal studies to investigate the possible role of FSH in osteoporosis prevention. The inhibition of the FSH effect successfully hindered osteoclast formation in-vitro. When injected in ovariectomized mice, the FSHβ antibody significantly attenuated bone loss by stimulating bone formation and inhibiting bone resorption [36]. Notably, this evidence addressed a role for FSH in the pathogenesis of bone loss. The implications of these findings in clinical practice need to be explored.

Scanty evidence is available on the impact of luteinizing-hormone (LH) on bone metabolism. LH receptors (LHR) are expressed in osteoblasts. In contrast to FSH, LHR knockout mice have lower bone mass compared to wild type animals, which is apparently secondary to the suppressed gonadal steroid production [37].

### 2.4. Parathyroid Hormone and Vitamin D

Parathyroid hormone (PTH) has a significant effect on bone metabolism, triggering both bone resorption and bone formation, depending on which cell-types are activated and the temporal pattern of activation [38]. In more detail, by stimulating the expression of RANKL and RANK and by inhibiting the secretion of OPG, it enhances osteoclastogenesis and bone resorption, mainly contributing to cortical bone loss [39]. PTH has also been suspected to play a role in bone modeling following mechanical stress. In particular, it has been hypothesized to induce bone formation by binding the PTH-related peptide type 1 receptor (PPR) on osteoblasts. Accordingly, conditional PPR-knockout mice (PPRcKO) (with a selective osteoblast PPR down-regulation) showed loss of bone mass under sedentary conditions. PPRcKO mice exposed to treadmill running showed significantly less structural-level and tissue-level mechanical properties when compared to wild-type, thus suggesting that PPR activation in osteoblasts is required for exercise to improve the mechanical properties of cortical bone [40].

Vitamin D has a protective role on bone mass, mainly acting on osteoblasts. Accordingly, mice with a constitutive knock out in the Vitamin Dreceptorgene have increased bone mass in radiological assessment [41]. In addition, the vitamin D receptor is expressed in both osteocytes and osteoblasts. In the latter, it mediates vitamin D-induced bone mass increase by suppressing bone resorption [42]. Indeed, the conditional knock out for this receptor in osteoblasts results in the failure of vitamin D administration to improve bone mass [42]. Finally, increased vitamin D levels have also been found to enhance bone formation by triggering osteoblast differentiation [42].

### 2.5. Insulin-Like Growth Factor 1

Osteoblasts express insulin-like growth factor 1 (IGF1) receptor (IGF1R) of which its activation enhances osteoblastogenesis by triggering the PI3K/Akt pathway. Accordingly, IGF1R stimulation in pre-osteoblasts stimulates type I collagen synthesis and osteogenic protein expression (ALT and OCN) in-vitro. Similarly, in-vivo data showed that IGF1 also has a role in bone matrix deposition [43]. Ex-vivo experiments on osteoclasts indicate that IGF1 enhances osteoclasts activity and bone resorption. The inactivation of the IGF axis in osteocytes compromises cortical bone morphology and suggests a role in periosteal bone formation [43]. In summary, IGF1 influences bone growth and modulates cortical and trabecular bone properties acting on osteoblast, osteocyte, and osteoclast function. Collectively, a protective role on bone mass has been suggested since low IGF1 levels have been implicated in bone loss [43].

## 3. Hormonal Changes in the Elderly

In this section, we will discuss the main hormonal changes that occur in the elderly. Table 2 summarizes these changes.

### 3.1. Thyroid Hormones and Thyroid-Stimulating Hormone

Recent data from observational studies suggest that serum TSH levels increase in older people. The Whickham survey was the first population-based study to evaluate the presence of thyroid dysfunction in community-dwelling individuals. This study reported that TSH levels increased with age in women older than 45 years, but not in men [44]. The larger and most recent US National Health and Nutritional Examination Survey (NHANES) III study showed that serum TSH and antibodies against thyroid peroxidase (TPOAb) and thyroglobulin (TgAb) increased with age in both men and women [45]. Moreover, in both the Cardiovascular Health Study (CVHS All Stars Surveys) and the Busselton survey study, a significant rise in TSH levels with little or no change in FT4 levels over a 13-year period of observation was found [46,47].

As TSH concentrations increase with age, the prevalence of subclinical hypothyroidism may be overestimated in the elderly. Indeed, reference ranges for TSH and thyroid hormones derive mainly from younger populations and age-specific ranges are not used in routine clinical practice. By using a uniform TSH reference range across all age groups in the NHANES study, Surks and Hollowell found that approximately 70% of older people with a slightly high serum TSH were incorrectly considered to have subclinical hypothyroidism [48]. Their analysis suggested that the 97.5 centile, corresponding to the upper limit of the TSH reference range, was about 3.6 IU/L in people who were 20–39 years of age and 5.9 and 7.5  IU/L in those who were 70–79 and 80 years old and older, respectively [48]. Thus, a significant number of older people may be inappropriately treated with thyroid hormones with the risk of subclinical hyperthyroidism and its clinical consequences. In the UK, it is estimated that hypothyroidism treated with levothyroxine may affect about 800,000 older individuals aged more than 70 years [49]. Hyperthyroidism is clearly less common than hypothyroidism: the NHANES study showed a prevalence of 1.3%, which includes subclinical and overt hyperthyroidism [45]. However, this same study showed a clear increase in the prevalence of hyperthyroidism during aging with a prevalence of 4–7% in people aged more than 70 years [45]. With aging, autonomously functioning thyroid nodules become the predominant cause of hyperthyroidism, especially in iodine-deficient areas [50]. Therefore, thyroid dysfunctions, both slightly elevated and slightly suppressed TSH, increase in the elderly.

The precise mechanisms underlying the alteration of thyroid function in older people are not entirely clear. TSH biological activity may decrease with age, possibly due to changes in TSH glycosylation or an age-related decrease in thyroid gland sensitivity to TSH [51]. However, more studies are needed to understand if these changes are part of healthy aging or are a bio-marker of underlying disease.

### 3.2. Glucocorticoids

Several changes occur in adrenal gland activity during aging. In a recent review, Yallouris and colleagues discussed adrenal aging and its clinical implications [52]. Adrenal gland changes with aging are associated with different hormonal output, i.e., an increase in GC secretion and a decline in adrenal androgen and aldosterone levels. The most significant modification is the increase in mean daily serum cortisol levels in the elderly [53,54]. The normal circadian rhythm pattern (i.e., cortisol peak in the early morning and cortisol nadir in the evening) is preserved [55,56] with an evening and night time higher nadir [57,58] and an attenuated awakening response and a lower early morning peak [58]. In addition, there is a reduction of hypothalamic-pituitary-adrenal (HPA) axis negative feedback, which could be associated with several factors, such as vascular components, a reduced number of brain GC receptors, differences of cortisol concentration in the cerebrospinal fluid (CSF), and alterations of cortisol clearance in the blood brain barrier or the CSF [59].

### 3.3. Sex Hormones and Gonadotropins

It is widely recognized that serum estrogens fall and gonadotropins rapidly rise after menopause as a result of primary ovary insufficiency in elderly women. Likewise, alterations of the testicular function occur in older men. Indeed, sex hormone levels undergo a slow decline in men during aging. Several large cross-sectional studies have shown an approximate 1% to 2% annual decline in T levels [60,61]. Circulating T is bound with high affinity to SHBG and albumin, and only 0.5–3% of T remains free, representing the biologically active fraction [62]. The concentration of SHBG increases with aging, which means that the concentration of free T decreases. Data from the Massachusetts Male Aging Study (MMAS) estimated the annual rate of total testosterone decline to be 0.4%, with free T of 1.2%, whereas SHBG increases by 1.2% [63]. A longitudinal study published by Travison and colleagues showed that serum total T decreases by 2.7%, while at the same time, SHBG increases by 2.7% in men who are 55–68 years old [64].

The combination of low T and hypogonadism-related symptoms in older men is defined “late-onset hypogonadism” (LOH) [65]. The European Male Ageing Study (EMAS) defined strict criteria for the diagnosis of LOH. They include the simultaneous presence of low serum T (total T < 11 nmol/L and free T < 220 pmol/L), which was confirmed at least twice, and three sexual symptoms (erectile dysfunction, decreased morning erections, and decreased sexual thoughts) [66]. Thus, when only biochemical criteria are used (i.e., T below the lower limit of the reference range of young men), the prevalence of hypogonadism is higher [67]. According to the strict EMAS criteria, the prevalence of LOH is about 2% in 40–80 year old men.

Interestingly, data from MMAS showed an increase of both LH and FSH, respectively, by 1.1% and 3.5% per year of age [60], thus confirming the primary nature of LOH. Accordingly, the total number of Sertoli and Leydig cells decreases to around half the number seen in the testis of a young man [68,69]. However, both obesity and chronic diseases (e.g., type 2 diabetes mellitus, metabolic syndrome, cardiovascular and chronic obstructive pulmonary disease, and frailty) have a significant role in the decrease of T secretion during aging [70]. Obesity as the cause of low T in elders is more frequent. EMAS showed that 73% of patients with LOH are obese or overweight [66]. This form of hypogonadism is characterized by low T with low or inappropriately normal serum LH levels (secondary hypogonadism).

### 3.4. Parathyroid Hormone and Vitamin D

The incidence of both primary and secondary hyperparathyroidism has been found to increase with age. Primary hyperparathyroidism (PHPT) increases up to 5:1 after the age of 75 years [71]. In most cases, PHPT is caused by a single parathyroid adenoma (75–85%), whereas 15–20% of cases are caused by hyperplasia affecting more than one parathyroid gland [71]. Secondary hyperparathyroidism is frequent both due to the high prevalence of vitamin D deficiency as well as due to decreased kidney function. Indeed, reduced calcium absorption due to vitamin D deficiency results in increased PTH secretion to maintain calcium homeostasis.

Vitamin D is mainly produced in the skin through the effect of the sun’s UVB on 7-dehydrocholesterol. Then, it requires two hydroxylations to become activated. The first one occurs in the liver by 25-hydroxylase (CYP2R1) with the production of 25-hydroxyvitamin D (25(OH)D), which is generally considered the best marker of vitamin D status. The second one mainly occurs in the kidney and consists of hydroxylation by 1 alpha-hydroxylase (CYP27B1) to 1,25 dihydroxyvitamin D (1,25 (OH) 2D). Vitamin D deficiency is a common finding among the elderly. Many factors influence the production and the activity of vitamin D during aging. First of all, elderly people spend less time outdoors, especially if they are institutionalized [72]. In addition, the dermal capacity to produce vitamin D in older people has been estimated to be about 25% of those aged 20–30 years when exposed to the same amount of sunlight [73]. This decrease seems to be related to the reduction in the concentration of 7-dehydrocholesterol in the skin [74]. Vitamin D deficiency may contribute to decrease calcium absorption. Indeed, intestinal calcium absorption decreases with advancing age [72,75] and the development of intestinal resistance to 1,25(OH)2D may contribute to the lower calcium absorption [76]. Moreover, renal function declines during aging and this is accompanied by an impaired hydroxylation of 25(OH)D to 1,25(OH)2D3 [75,77]. Concomitantly with the decline in renal production of 1,25(OH)2D, there is also an age-related decrease in renal vitamin D receptor (VDR) and epithelial calcium (Ca^2+^) channels TRPV5 expression, which lowers renal calcium reabsorption efficacy [78].

### 3.5. Insulin-Like Growth Factor 1

Serum IGF1 has long been known to decline with age [79]. The English Longitudinal Study of Ageing, carried out in patients older than 50 years, reported higher mean IGF-1 values in men than women and, as expected, the occurrence of an inverse correlation with age in both sexes [80].

## 4. Management of Osteoporosis: Novel Possible Targets

In view of the influence that many hormones have on bone metabolism, their changes during aging must be taken into account to better understand the pathogenesis of primary osteoporosis in the aging population and the endocrine panorama should not be restricted to sex hormone variations. Indeed, the drastic decrease of estrogens following menopause has historically been implicated in female osteoporosis. In line with this rationale, selective estrogen receptor modulators, stimulating the ER in bone tissue, are currently used for the treatment of osteoporosis [76]. However, other hormones should be considered at the same time, particularly some that act to prevent bone loss. This is the case of TSH, which prevents bone loss and of which its levels increase in the elderly. On the other hand, the fluctuation that other hormones have in aging people may favor bone loss. With this rational, suppression of TSH by L-thyroxine administration for the treatment of thyroid nodules should be avoided in the elderly [81,82].

Cortisol, by hampering osteoblastogenesis and bone formation, increases in the elderly. The excess of cortisol during aging contributes to BMD decline, leading to osteopenia, osteoporosis, and increased risk of fractures through the stimulation of osteoblast and osteocyte apoptosis [83], increased osteoclast survival, and the suppression of new osteoblast formulation [58].

Testosterone, vitamin D, and IGF1 have been reported to decline in old people, thus contributing to the decrease of bone formation. The negative impact that testosterone or vitamin D deficiency have on BMD is widely recognized. Indeed, the scientific societies have developed guidelines on the management of osteoporosis in hypogonadal men, and testosterone as well as vitamin D have been included among the therapeutic choices [76,84] (Table 3). Less recognized is the role of IGF1 in bone metabolism. A few clinical studies have associated the IGF1 decline in the elderly with the decrease of BMD. The Framingham Osteoporosis Study, carried out in men and women aged 72–74, reported that high IGF1 levels are associated with greater BMD in older women [85]. In addition, low IGF1 levels are associated with increased fracture risk [86,87]. Other studies do not confirm this evidence and trials with rhGH or rhIGF1 in older patients with primary osteoporosis or frailty led to conflicting results [43].

Increasing interest has grown with respect to the role of FSH in bone metabolism in recent years. As preclinical data indicate, FSH enhances osteoclastogenesis and favors bone loss. Therefore, the rise in FSH levels that is observed in post-menopausal women has been addressed in the pathogenesis of osteoporosis [88]. Accordingly, the AA rs6166 FSHR genotype, encoding for a more sensitive FSHR than that encoded by the GG one, is associated with low total body BMD, independently of circulating estrogen [89]. The data have been confirmed by lower femoral neck, calcaneal, and total-body T-scores in post-menopausal women with the AA rs6166 FSHR compared to the GG genotype [90]. Such evidence may also pertain to the male sex, where an FSH increase also occurs in the elderly. A case-control study performed in older men revealed that FSH was a negative predictor of BMD at lumbar spine, femoral neck, and hip in the adjusted multivariate model [91]. Similarly, according to longitudinal data from the 5-year-long Concord Health and Ageing in Men Project in 1705, men aged >70 years reported a negative association between FSH values and BMD, independently of testosterone levels [92], thus confirming the negative impact of FSH on bone health. Interestingly, high FSH serum levels occur in Klinefelter syndrome (KS), a syndrome with a prevalence of about 1:660 men [93]. In these men, bone loss has long been considered due to testosterone deficiency, despite the fact that a role of FSH increase could not be excluded. However, no data are currently available on this topic.

Interestingly, a polyclonal antibody with a FSHR-binding sequence against the β-subunit of murine FSH has been developed. When administered to ovariectomized mice, it was effective in ameliorating bone loss [36]. Immunotherapy is already used for the treatment of osteoporosis as denosumab, a humanized monoclonal antibody that neutralizes RANKL, which is a cytokine that induces osteoclastogenesis by binding to its receptor (RANK), is an available therapeutic choice both for male and female osteoporosis [76,84]. Hence, the development and the use of a monoclonal antibody against the FSHβ chain might be hypothesized for the treatment of osteoporosis. In addition, the rs6166 FSHR genotype might be gathered as a diagnostic tool to stratify the risk of developing osteoporosis in the female sex [89,90]. The role of this polymorphism in male osteoporosis (including KS patients) deserves further studies to be explored more deeply.

However, it must be considered that several other factors might also influence bone metabolism. Accordingly, chronic inflammation, often occurring during aging, has been shown to promote bone loss [94] and an increased level of interleukine-31, a proinflammatory cytokine, which has been observed in post-menopausal women with decreased BMD [95].

## 5. Conclusions

In conclusion, while the rise in TSH levels has a protective role on bone mass, the decline of estrogen, testosterone, IGF1, and vitamin D and the rise of cortisol, parathyroid hormone, and FSH favor bone loss in the elderly. Particularly, the estrogen serum level decrease and the consequent FSH increase enhance bone resorption, whereas the decline of testosterone and IGF1 negatively impact on bone formation. Despite the fact that a direct effect of vitamin D on osteoblastogenesis has been reported, it mainly impacts bone metabolism due to the rise of parathyroid hormone levels, which, in turn, induces osteoclastogenesis. Hence, a complete hormonal assessment should be completed for both women and men during bone loss evaluation. Furthermore, novel possible diagnostic and therapeutic tools might be developed for the management of osteoporosis. In fact, the rs6166 FSHR genotype might help in stratifying the risk of developing osteoporosis and a monoclonal antibody against the FSHβ chain has shown promising results for the treatment of osteoporosis in the animal model [36]. Finally, studies aimed at assessing the difference in terms of BMD in hypogonadotropic and hypergonadotropic hypogonadal patients should be carried out.

## Figures and Tables

**Table 1 jcm-08-01564-t001:** The role of hormones on bone mass.

Hormones	Molecular Action	Role
TSH	↑osteoblast differentiation ↓osteoclast formation and survival	+
Cortisol	↓maturation, lifespan, and function of osteoblast	−
Estradiol	↑osteoblast proliferation and differentiation ↓osteoclast differentiation	+
Testosterone	↑osteoblast proliferation and differentiation	+
FSH	↑osteoblast proliferation	−
LH	Unclear	Unclear
Parathyroid hormone	↑osteoclast proliferation ↑osteoblast proliferation	−
Vitamin D	↑osteoblast differentiation	+
IGF1	↑osteoblast proliferation and differentiation ↑osteoblast proliferation	+

Abbreviations: FSH = follicle-stimulating hormone; IGF1 = insulin-like growth factor 1; LH = luteinizing hormone; TSH = thyroid stimulating hormone. +, preventing role on bone loss; −, favoring role on bone loss.

**Table 2 jcm-08-01564-t002:** The main hormonal changes in the elderly.

Hormones	Changes during Aging
TSH	↑
Cortisol	↑
17ß-Estradiol	↓
Testosterone	↓
FSH	↑
LH	↑
Vitamin D	↓
Parathyroid hormone	↑
IGF1	↓

Abbreviations: FSH = follicle-stimulating hormone; IGF1 = insulin-like growth factor 1; LH = luteinizing hormone; TSH = thyroid stimulating hormone.

**Table 3 jcm-08-01564-t003:** Drugs for the treatment of osteoporosis.

Drug	Dose	Route of Administration	Side Effects	Indications
Alendronate	10 mg/day or 70 mg/week	Oral	Atypical femoral fracture, osteonecrosis of the jaw, gastrointestinal symptoms, muscle and joint pain	Treatment of osteoporosis in men and women
Risedronate	5 mg/day or 35 mg/weekly or 75 mg on 2 consecutive days once a month	Oral	Atypical femoral fracture, osteonecrosis of the jaw, gastrointestinal symptoms, muscle and joint pain	Treatment of osteoporosis in men and women
Zoledronic acid	5 mg every 12 months	IV (intravenous)	Atypical femoral fracture, osteonecrosis of the jaw, gastrointestinal symptoms, influenza-like symptoms, hypocalcemia	Treatment of osteoporosis in men at increased risk of fracture and women
Denosumab	60 mg every 6 months	SC (subcutaneous)	Atypical femoral fracture, osteonecrosis of the jaw, hypocalcemia, hypersensivity reactions	Treatment of osteoporosis in men and women. It might be the first option in the case of renal failure and high risk of fractures, and after failure or adverse events of other treatments
Teriparatide	20 or 40 µg/day	SC	Gastrointestinal symptoms, headache, dizziness, muscle pain, hypercalcemia, hypercalciuria, renal side effects	Severe osteoporosis at increased risk of fracture in patients who experience a new spine or hip fracture after 1 year of treatment with other anti-resorptive drugs
Strontium Ranelate	2 g/day	Oral	Increased risk for tromboembolic events and myocardial infarction, allergic reactions	Adult patients at high risk of fracture, for whom treatment with other drugs approved for the osteoporosis is not possible
Testosterone	Minimal necessary dose to maintain T serum concentrations in the middle tertile of the normal physiological range	Various formulations		Male hypogonadism
SERMs (raloxifene)	60 mg/day	Oral	Hot flushes, leg cramps, increased risk for thromboembolic events	Women with a low risk of deep vein thrombosis and for whom bisphosphonates or denosumab are not appropriate, or with a high risk of breast cancer
Tibolone	1.25 mg/day	Oral	Stroke, vaginal discharge, and bleeding	Women under 60 years of age or 10 years after menopause at high risk of fractures with climacteric symptoms
Estrogen with or without progestogen	Oral conjugated equine estrogen: 0.625 mg/day; estradiol: 100 mg patch or 2 mg/day orally	Oral or transdermal	Venous thromboembolism, stroke, myocardial infarction, cancer (breast, endometrial, ovary), dementia, gallbladder disease, and urinary incontinence	Postmenopausal women (under 60 years of age or 10 years past menopause) at high risk of fracture (estrogens are suggested only in women with hysterectomy)

SERMs: Selective estrogen receptor modulators.

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
