# Peer review of "Osteoporosis from an Endocrine Perspective: The Role of Hormonal Changes in the Elderly"

_jcm, 2019, doi:10.3390/jcm8101564_

Round 1

Reviewer 1 Report

This is a review of endocrine effects on bone health in the elderly, based on a Medline search.  The paper is quite informative and readable, but please see the following comments/suggestions:

General:  It may be preferable to use "sex" rather than "gender" throughout, since biological sex is the relevant variable.

Line 46:  Modelling is important during growth and development, not just after mechanical stress (although these are often related).

Line 52:  It is not clear what is meant by an increased modeling rate resulting in a negative balance--can the authors provide a reference?

Line 57:  Grammatical comment--"poor" should be "poorly"

Line 58-61:  How many articles were identified for this review?

Line 174:  It is not clear why the radius is the only bone that is discussed here with regard to cortical bone, which is found throughout the skeleton.

Line 294:  Raloxifene is a SERM that is already approved (in the U.S.) for the prevention and treatment of osteoporosis in post-menopausal women, i.e. it is not clear why the word "proposed" is used.  Is it referring to other SERMS that are not yet on the market?

Line 331 (paragraph): It is not clear why there is so much discussion about the polyclonal antibody with a FSHR-binding sequence.  This also appears in the abstract and conclusion. This reviewer is not a basic researcher but it seems that there is only pre-clinical data on this and perhaps it is not necessary to include it in a clinical review.

Table 3:  Why is there a table of approved drugs for men but not women? Adding this might improve the utility of the review for clinicians.

Author Response

Manuscript ID jcm-591714 Revised

Comment 1: General:  It may be preferable to use “sex” rather than “gender” throughout, since biological sex is the relevant variable.

Answer to Comment 1: We changed “gender” with “sex” throughout the text, as suggested.

Comment 2: Line 46:  Modelling is important during growth and development, not just after mechanical stress (although these are often related).

Answer to Comment 2: We agree and, accordingly,the sentence in lines41-45 was rephrased.

Comment 3: Line 52:  It is not clear what is meant by an increased modeling rate resulting in a negative balance--can the authors provide a reference?

Answer to Comment 3: The reference is the n. 7 (Osteoporosis. Lancet 2019, 393(10169), 364-376), which provides an overview of the management and treatment of osteoporosis. We sought to express that the pathogenesis of osteoporosis in male and female sex is different, since it is mainly due to an excessive bone reabsorption in the female sex and a lack of bone formation in the male sex. We rephrased the sentence you referred to, so to make this concept clearer (please see line 49).

Comment 4:Line 57:  Grammatical comment - “poor” should be "poorly"

Answer to Comment 4: Corrected. Thank you.

Comment 5:Line 58-61:  How many articles were identified for this review?

Answer to Comment 5:Eighty-two articles have been included in the qualitative synthesis (references 8-90).

Comment 6:Line 174:  It is not clear why the radius is the only bone that is discussed here with regard to cortical bone, which is found throughout the skeleton.

Answer to Comment 6: We agree with your comment. We deleted the discussion on “radius”.

Comment 7:Line 294:  Raloxifene is a SERM that is already approved (in the U.S.) for the prevention and treatment of osteoporosis in post-menopausal women, i.e. it is not clear why the word “proposed” is used.  Is it referring to other SERMS that are not yet on the market?

Answer to Comment 7: The term “proposed” has been improperly used.Line 297 has been rephrased.

Comment 8:Line 331 (paragraph): It is not clear why there is so much discussion about the polyclonal antibody with a FSHR-binding sequence. This also appears in the abstract and conclusion. This reviewer is not a basic researcher but it seems that there is only pre-clinical data on this and perhaps it is not necessary to include it in a clinical review.

Answer to Comment 8: Thank you for your comment. The majority of data on the role of FSH on bone metabolism arepreclinical. The findings of a clinical study carried out in post-menopausal women seem to confirm preclinical findings (please see ref. n. 91). We sought to highlight this evidence since we believe it might represent a promising target for the management of osteoporosis in the future.

Comment 9: Table 3:  Why is there a table of approved drugs for men but not women? Adding this might improve the utility of the review for clinicians.

Answer to Comment 9: We appreciated your comment.Such information has been included in revised Table 3.

Reviewer 2 Report

General comment

This paper describes first, in the introduction, osteoporosis in general, which seems inadequate to his reviewer.

The chapter "preclinical evidence" present data on the hormonal  influences on bone cells and bone metabolism, but not specifically related to advanced age as indicated in the title o the paper.

In the chapter "hormonal changes in the elderly" there is a detailed description of the the endocrinology in the elderly,  but in relation to bone.

Finally, on the chapter "management of osteoporosis" there is no presentation of the literature on this topic, but a description of the bone effects of hormones in the elderly. This finally corresponds to the title of the paper, although it is not treating "management of osteoporosis" an announced.

There is a major discrepancy between the content of the paper. In addition, the content deals partially with bone, but only in preclinical research, and not in clinical research. Then it skips to to general endocrinology in the elderly - not specifically related to bone, and finally comes to the announced topic in the last chapter, "management". Here are described the bone effects of various hormones in the elderly, but without treating the management (except the table on drugs, which is out of context). There is e.g. a literature on therapeutical trials with testosterone. 

 Specific comments:

The introduction on osteoporosis in general must be left out ( a linealinea 50-53).

al 166: the effect on Vitamin D deficiency is lacking.

al 184-283: does not discuss any bone effect

al 289: hormones have also an effect on bone formation, not only on bone loss.

Author Response

Manuscript ID jcm-591714 Revised

Comment 1: This paper describes first, in the introduction, osteoporosis in general, which seems inadequate to this reviewer.

Answer to Comment 1: Thank you for you comment. We shortened the Introduction section, hoping that is now more it now more appropriate for this review. We think that a brief paragraph introducingosteoporosis, the role of osteoblasts and osteoclasts and hormonal influence on bone metabolism is useful for the reader.

Comment 2:The chapter “preclinical evidence” present data on the hormonal influences on bone cells and bone metabolism, but not specifically related to advanced age as indicated in the title of the paper.

Answer to Comment 2: The aim of the paragraph “preclinical evidence” was to show data on the influence of different hormones in bone metabolism (please see lines 52-54).

Comment 3:In the chapter “hormonal changes in the elderly” there is a detailed description of the endocrinology in the elderly, but in relation to bone.

Answer to Comment 3: The aim of the paragraph “hormonal changes in the elderly” was briefly summarizing the evidence showing that changes in the hormonal serum levels occurin the elderly. Please see the aims of this review in lines 52-54.

Comment 4:Finally, on the chapter “management of osteoporosis” there is no presentation of the literature on this topic, but a description of the bone effects of hormones in the elderly. This finally corresponds to the title of the paper, although it is not treating “management of osteoporosis” as announced.

Answer to Comment 4: The paragraph “Management of osteoporosis: novel possible targets” has the purpose of matching the evidence discussed in the previous two paragraphs. Accordingly, since hormonal changes occur in the elderly (Table 2) and considering the impact of these hormones on bone metabolisms (Table 1), novel possible targets might be identified in the future management of osteoporosis. Particularly, we believe that FSH might represent a promising target for the management of osteoporosis in the future.

Comment 5: There is a major discrepancy between the content of the paper. In addition, the content deals partially with bone, but only in preclinical research, and not in clinical research. Then it skips to general endocrinology in the elderly - not specifically related to bone, and finally comes to the announced topic in the last chapter, “management”. Here are described the bone effects of various hormones in the elderly, but without treating the management (except the table on drugs, which is out of context). There is e.g. a literature on therapeutical trials with testosterone.

Answer to Comment 5: Thank you for your comment. A great body of evidence is currently available on the management of osteoporosis and therapeutic options. This review attempted to suggest a possible novel interpretation of the pathogenesis of osteoporosis in the elderly. As explained in the answers to comments 2-4, this review gathers together preclinical evidence showing the impact of hormones on bone metabolism with clinical evidence showing that several hormonal changes occur in the elderly. Altogether these data suggest that the pathogenesis of osteoporosis in the elderly may be at least partly due to thedecline of estrogen, testosterone, IGF1 and vitamin D and to the rise of cortisol, parathyroid hormone and FSH. Therefore, according to an endocrinological point-of-view, the assessment of such hormones might be taken into account in the pathogenesis of osteoporosis.

Comment 6: The introduction on osteoporosis in general must be left out (lines50-53).

Answer to Comment 6:The lines you refer to (“Osteoporosis-associated fractures are increasingly prevalent in women after 55 years of age, as well as in men after 65 years of age, indicating the negative impact of aging on bone metabolism. Aging affects the remodeling balance in a gender-specific manner. In women, it is associated with an increased bone modelling rate, resulting in a negative balance. In men, aging is mostly associated with reduced bone formation and low bone turnover”) have been rephrased, according to the Reviewer 1 suggestions. These lines are important to introduce the reader into the aim of the review (Please see lines 49-54).

Comment 7:al 166: the effect on Vitamin D deficiency is lacking.

Answer to Comment 7: Thank you for this comment. We have extended this section (lines 162-167).

Comment 8:al 184-283: does not discuss any bone effect

Answer to Comment 8: Please see Answer to Comment 2.

Comment 9:al 289: hormones have also an effect on bone formation, not only on bone loss.

Answer to Comment 9:We have replaced “loss” with “metabolism”.

Reviewer 3 Report

The manuscript is interesting and well thought-out.

The aim of the review is to examine the evidence of hormone influence on bone metabolism.

However I would like the authors to:

Add some hints about the pathogenesis of osteoporosis and the multifactorial role of inflammation, several cytokines and emerging therapies (- Ciccarelli et al. Glucocorticoids in Patients with Rheumatic Diseases: Friends or Enemies of Bone? Curr Med Chem. 2015; 22(5):596-603) Ginaldi et al. Increased levels of interleukin 31 (IL-31) in osteoporosis. BMC Immunol. 2015 Oct 8;16:60) Ginaldi et al. Interleukin-33 serum levels in postmenopausal women with osteoporosis. Sci Rep. 2019; doi: 10.1038/s41598-019-40212-6 De Martinis M et al. Osteoporosis: Current and emerging therapies targeted to immunological checkpoints. Curr Med Chem. 2019 Jul 30. doi: 10.2174/0929867326666190730113123. Ilesanmi-Oyelere BL et al. Inflammatory markers and bone health in postmenopausal women: a cross-sectional overview. Immun Ageing. 2019 Jul 10;16:15. doi: 10.1186/s12979-019-0155-x. eCollection 2019. ).
Update references.

Author Response

Manuscript ID jcm-591714 Revised

Comment 1:I would like the authors to:Add some hints about the pathogenesis of osteoporosis and the multifactorial role of inflammation, several cytokines and emerging therapies (- Ciccarelli et al. Glucocorticoids in Patients with Rheumatic Diseases: Friends or Enemies of Bone? Curr Med Chem. 2015; 22(5):596-603) Ginaldi et al. Increased levels of interleukin 31 (IL-31) in osteoporosis. BMC Immunol. 2015 Oct 8;16:60) Ginaldi et al. Interleukin-33 serum levels in postmenopausal women with osteoporosis. Sci Rep. 2019; doi: 10.1038/s41598-019-40212-6 De Martinis M et al. Osteoporosis: Current and emerging therapies targeted to immunological checkpoints. Curr Med Chem. 2019 Jul 30. doi: 10.2174/0929867326666190730113123. Ilesanmi-Oyelere BL et al. Inflammatory markers and bone health in postmenopausal women: a cross-sectional overview. Immun Ageing. 2019 Jul 10;16:15. doi: 10.1186/s12979-019-0155-x. eCollection 2019. ).

Answer to comment 1:Done, as suggested. Please see lines 345-348.

Comment 2:Update references.

Answer to comment 2:We appreciated your suggestion. We have included not recent information since the major of preclinical evidence showing the influence of hormones on bone loss is dated.